# Enzyme-Assisted Extraction for the Recovery of Food-Grade Chlorophyll-Based Green Colorant

**DOI:** 10.3390/foods12183440

**Published:** 2023-09-15

**Authors:** Caterina Mazzocchi, Ilaria Benucci, Claudio Lombardelli, Marco Esti

**Affiliations:** Department of Agriculture and Forest Sciences (DAFNE), Tuscia University, Via S. Camillo de Lellis snc, 01100 Viterbo, Italy; caterina.mazzocchi@unitus.it (C.M.); ilaria.be@unitus.it (I.B.); esti@unitus.it (M.E.)

**Keywords:** solvent-free extraction, circular economy, zinc, natural pigment recovery, response surface methodology

## Abstract

The aim of the study was to develop a biotechnological approach for the green recovery of chlorophyll from spinach, to be used as a natural food colorant. The plant matrix was characterized in terms of cell wall polysaccharide composition, and a tailored enzymatic mix based on cellulase (40%) xylanase (41%) and polygalacturonase (19%) was formulated. The process variables (temperature (°C), time (h), enzyme mix dose (U/g), zinc concentration (ppm), and buffer/substrate ratio (B/S)) and their interactions were studied by response surface methodology. The overlay plot made it possible to identify the process conditions (T: 25 °C, Zn: 150 ppm e B/S: 17.5, t: <2 h and enzyme mix dose between 12 and 45 U/g) to maximize the amount of chlorophyll, and concurrently, the quality of the green color of the extract. Finally, the novel colorant was applied in the production of a real food.

## 1. Introduction

It has been estimated that about 46 tons of food per year remain unsold on the shelves of large-scale retail trade, and only a small percentage of that is intended for food banks and charity organizations [1]. In order to better manage food loss and food waste, the European Union adopted a Directive in 2018 [2] favoring the circular economy (CE) model. Indeed, the CE may contribute to food waste reduction through valorization in accordance with the principles of reusing, reducing, and recycling [3,4]. In this scenario, CE encourages the use of unsold food (especially fruit and vegetables) as a cheap and valuable source for the recovery of high-added-value molecules [5].

In this context, there is a growing interest in the extraction of colorants from natural sources [6,7,8,9]. Among natural pigments, chlorophyll (Chl) is widely distributed in fruits and vegetables (also in algae and cyanobacteria) and may be used as an alternative to synthetic dyes. The following green colorants are currently available on the market, all recovered using organic solvents: E140(i) chlorophyll extracted from plants; E140(ii) chlorophyllins, obtained by saponification of E140(i); E141(i) copper chlorophylls, obtained by substitution of the magnesium (Mg^2+^) with the copper (Cu^2+^) ion; and E141(ii) copper chlorophyllins [10]. The main sources of Chl are leafy vegetables such as spinach (*Spinacia oleracea*), lettuce (*Lactuca sativa*), and broccoli (*Brassica oleracea*), among others [11]. In detail, spinach is a highly consumed vegetable worldwide, and its production led to 25% of waste, from which high-added-value molecules can be extracted. In fact, spinach by-products may contain high amounts of lutein (3.9–9.5 mg/100 g_FW_) and chlorophyll (about 130 mg/100 g_FW_). Recently, these extracts have been used to develop nutraceuticals or functional food products [12,13,14,15,16].

Chl is an oil-soluble pigment and occurs in nature in two different structures: chlorophyll-a (blue-green color), if in the 7-carbon position it has a methyl group, and chlorophyll-b (blue-yellow color), if it shows an aldehyde group in the same position [11,17]. Chl is a stable pigment in nature, but factors such as oxygen, pH, light, temperature, and chlorophyllase enzymes may lead to degradation phenomena when the pigment is extracted from its natural environment [18]. All these factors may cause a centralized shift of the Mg^2+^ ion in the central position, giving rise to pheophytin, which causes a color change towards an olive green/gray tone. It is possible to produce more stable complexes by substituting Mg^2+^ with Cu^2+^ or Zn^2+^, thus restoring the functions and properties of the molecule and producing a greening effect [19]. Although Cu^2+^ is more easily inserted into the structure of the molecule than Zn^2+^, the toxicity of Cu^2+^ is greater than that of Zn^2+^ [7].

Commonly, Chl is recovered from plant matrices through various conventional methods such as maceration and Soxhlet extraction techniques with organic or inorganic solvents. These non-environmentally friendly and cost-intensive methodologies have several disadvantages, including low extraction yield and long process times [11]. For these reasons, “green” extraction techniques have been developed (such as supercritical fluid extraction, ultrasound assisted extraction, microwave assisted extraction, pulsed-electric field-assisted extraction, and enzyme-assisted extraction) [20,21,22]. Enzyme-assisted extraction (EAE) of pigments is based on the breakdown of the plant cell wall by enzymatic hydrolysis [19,23]. Cell-wall-degrading enzymes have been used successfully for the recovery of a variety of constituents, including anthocyanins [24], lycopene [25], β-carotene [26], carotenoids [4] and betalains [27] from plant tissues. Only a few studies investigated the application of EAE for the recovery of Chl via commercial standard preparations [7,28]. Özkan et al. [7] described the extraction of chlorophyll from spinach using Pectinex Ultra SP-L, which was not able to completely hydrolyze the plant cell wall. This is likely because the provided enzymatic activities are not flexible in relation to the vegetable tissue composition.

Therefore, in this study, a tailor-made EAE protocol has been formulated for the recovery of Chl from unsold spinach, considering its polysaccharide cell wall composition and concurrently optimizing the process parameters (such as temperature, time, total enzyme mix dose, zinc concentration, and buffer/substrate ratio) to maximize the amount of Chl extracted. Finally, the green Chl-based extract obtained by EAE was incorporated in the formulation of a real food (Baghrir) to evaluate its tinting strength.

## 2. Materials and Methods

### 2.1. Plant Material, Enzymes and Chemicals

Unsold spinach (*Spinacia oleracea*) was supplied by Unicoop Tirreno S.C. (Viterbo, Lazio, Italy) and stored at 4 °C until use.

The enzyme preparations (Merck, Milan, Italy) used were as follows: cellulase (CL, 0.8 U/mg) from *Aspergillus niger*, xylanase (XL, ≥2500 units/mg) from *Aspergillus oryzae*, and polygalacturonase (PG, ≥0.3 U/mg) from *Aspergillus niger*. All the reagents used, including enzyme substrates (polygalacturonic acid, carboxymethylcellulose, and xylan), were purchased from Merck (Milan, Italy).

### 2.2. Plant Material Characterization

#### 2.2.1. Moisture and Dry Matter

Moisture content (%) and dry matter (DM, %) were evaluated after drying spinach samples (100 g) in an air-forced oven (Venticell, MMM Medcenter Einrichtungen GmbH, Planegg/München, Germany) at 30 °C for 5 days [29].

DM was determined through the following equation (Equation (1)):DM = d_w_/f_w_ × 100(1)
where: DM: dry matter, f_w_: fresh weight (g), d_w_: dry weight (g).

#### 2.2.2. pH

The pH was determined according to the Official Methods of Analysis of AOAC INTERNATIONAL [30] by means of a pH-meter (HANNA Instruments, Woonsocket, RI, USA) after blending spinach with 20% distilled water (spinach/water).

#### 2.2.3. Cell Wall Polysaccharides Composition

The content of cellulose, hemicellulose (xylans), and pectin was determined as described by Ververis et al. [29] and Lombardelli et al. [4]. Sugars released following acid hydrolysis (0.7 g d_w_ sample boiled with 5 mL H_2_SO_4_ solution 72% *v*/*v* for 4 h) were determined by using enzymatic kits (Megazyme International, Bray, Ireland) of D-fructose/D-glucose, D-xylose and D-glucuronic acid/D-galacturonic acid. Carbohydrate concentration was obtained by means of appropriate calibration curves.

### 2.3. Characterization of Enzyme Preparations

Enzyme preparations (CL, XL and PG) were characterized in terms of protein content, electrophoretic profile, and kinetic parameters. In detail, protein content was determined using the Bradford method [31], and the Abs was spectrophotometrically detected at 595 nm (Shimadzu UV 2450, Milan, Italy). The calibration curve was obtained using Bovin Serum Albumin (BSA) as standard. The electrophoretic profile was analyzed by SDS-PAGE gel electrophoresis under reducing conditions as reported by Lombardelli et al. [4].

The activities of CL, XL, and PG were determined in 0.1 M McIlvaine buffer, pH 5.0 at 45 °C. The resulting reaction products (glucose, xylose, and galacturonic acid) were detected by the 3′,5′-dinitrosalicylic method (DNS), reading the Abs at 530 nm [32]. The specific activity (S.A.) was indicated as I.U./mg protein expressed as BSA_eq_ (I.U./mg_BSAeq_).

For each enzyme, the kinetic characterization was carried out under the test conditions described above, varying the substrate concentration for each enzyme preparation: CL (carboxymethylcellulose: 1.55–12.75 mg/mL), XL (xylan: 0.066–0.1 mg/mL), and PG (polygalacturonic acid: 0.2–1 mg/mL). The Michaelis-Menten equation was applied for the estimation of kinetic parameters V_max_ (I.U./mg_BSAeq_) and K_M_ (mg/mL) using a nonlinear regression procedure (GraphPad Prism 5.01, GraphPad Software, Inc., La Jolla, CA, USA.). Furthermore, the catalytic constant (k_cat_, min^−^^1^) and the affinity constant (K_a_, min^−^^1^/mg mL^−^^1^) were calculated.

### 2.4. Extraction Process Optimization—Response Surface Methodology

The process conditions (temperature (T, °C), time (t, h), total enzyme dose (enzyme mix, U/g), ZnCl_2_ concentration (Zn, ppm) and buffer/substrate ratio (B/S)) were optimized for maximizing the amount of recovered Chl (μg/g). An experimental design based on the statistical model of Design of Experiment (DOE) was used [4]. The variance for each factor evaluated was divided into linear, quadratic, and interactive components, and was represented using the following polynomial function (Equation (2)):y = b_0_ + b_1_x_1_ + b_2_x_2_ + b_3_x_3_ + b_4_x_4_ + b_5_x_5_ + b_11_x_1_^2^ + b_22_x_2_^2^ + b_33_x_3_^2^ + b_44_x_4_^2^ + b_55_x_5_^2^ + b_12_x_1_x_2_ + b_13_x_1_x_3_ + b_14_x_1_x_4_ + b_15_x_1_x_5_ + b_23_x_2_x_3_ + b_24_x_2_x_4_ + b_25_x_2_x_5_ + b_34_x_3_x_4_ + b_35_x_3_x_5_ + b_45_x_4_x_5_
(2)
where, the dependent variable (y) represents the amount of extracted Chl. The coefficients of the polynomial are represented by b_0_ (constant term), b_1_, b_2_, b_3_, b_4_, b_5_ (linear coefficients), b_11_, b_22_, b_33_, b_44_, b_55_ (square coefficients), b_12_, b_13_, b_14_, b_15_, b_23_, b_24_, b_25_, b_34_, b_35_, b_45_ (interacting coefficients). The significance of all terms of the polynomial function was statistically evaluated using the F-value with a probability (p) of 0.001, 0.01 or 0.05. Regression coefficients were then used to generate contour plots and overlay plots [4,33]. DOE and statistical analysis were performed using Minitab 17.1 software (Minitab Inc., State College, PA, USA).

The response surface methodology (RSM) was used to verify the presence of combined effects between the different variables and to determine a range of optimal conditions to maximize the amount of extracted Chl. Each variable has three levels: −1, 0, and +1. A total of 32 combinations, including three replicas of the central point, were performed in random order according to the central composite design [34]. The levels of the variables x1, x2, x3, x4, x5 and Χ1, X2 X3, X4, X5 are shown in Table 1.

### 2.5. Chlorophyll Recovery by Enzyme Assisted Extraction

Chl extraction from unsold spinach was performed as proposed by Özkan et al. [7] with some modifications:

**Pretreatment and homogenization of spinach leaves**. Whole spinach leaves (25 g) were blanched for 5 s at 100 °C to inactivate the endogenous chlorophyllases which could lead to Chl degradation. Subsequently, the leaves were homogenized with a mixer (Phillip HR 2068 blender) for 1 min at maximum speed, using part of the buffer. The ratio of biomass (spinach) and McIlvaine buffer (pH 5; 0.1 M) was varied according to the experimental plan (Table 1).

**EAE treatment**. The homogenized spinach was resuspended in 0.1 M McIlvaine buffer (pH 5) and the treatments were carried out following DOE reported in Table 1. In detail, the value ranges for each variable were: T = 25–50 °C, t = 1–5 h, total enzyme dose = 10–50 U/g, Zn = 0–300 ppm, and B/S = 5–30. Each range has been identified considering the conditions reported in the literature for pigment extraction [7,28]. The reaction mixture was kept under continuous stirring in the dark in a bioreactor (1 L), which was equipped with an external water jacket for temperature control and connected to a thermostat (MPM Instrument, Type M 900-TI, Bernareggio, Italy). After incubation, the homogenate was centrifuged (Heraeus Megafuge 16R Centrifuge, Thermo Scientific, Milan, Italy) at 4500 rpm for 5 min at 10 °C and the completely clear supernatant was discarded.

### 2.6. Chlorophyll Content and Colorimetric Characterization

An amount of 1 g of pellet was added to 10 mL of absolute ethanol. After stirring, the solution was centrifuged at 4500 rpm for 5 min at 10 °C. The recovered supernatant was used for the spectrophotometric measurement (UV-visible, Shimadzu UV 2450, Milan, Italy), recording the spectra (from λ 230 to 700 nm) using a 1 mm quartz cuvette (Hellma, Milan, Italy). The amount of Chl was determined through the following equation, obtained via modifications of the equations of Özkan et al. [7] and Lichtenthaler [35] (Equation (3)).
Chl (µg/g) = (5.24 × Abs_664_ + 22.24 × Abs_649_) × DF(3)
where: Abs_664_ = Absorbance at λ 664 nm, Abs_649_ = Absorbance at λ 649 nm, DF = Dilution factor.

The supernatant was also used for colorimetric characterization. Color measurements were performed using a CR-5 colorimeter (Konica Minolta, Tokyo, Japan) with the CIELa*b* system. The colorimetric analyses were performed in triplicate, with five measurements for each sample unit. Furthermore, extracts were evaluated through two derived color parameters: hue angle (h°) and chroma (C*) [26].

### 2.7. Application of Chlorophyll-Based Colorant in Real Food

After the EAE, the pellet containing Chl was freeze-dried (bench freeze-dryer, Labconco Corporation, Kansas City, MI, USA) and used for the application in a typical Moroccan dish: “Baghrir”. It is a sweet made with semolina (23% *w*/*w*), flour (9.20% *w*/*w*), sugar (1.74% *w*/*w*), salt (0.57% *w*/*w*), dry yeast (0.34% *w*/*w*), chemical yeast (1.84% *w*/*w*), and water (63.31% w/w). Chl-based extract was added to the Baghrir at different concentrations (1%, 2.5%, 5%, 7.5 and 10% *w*/*w* fluor based). An amount of 10 g of dough was cooked on a pan over medium heat until bubbles appeared on one side only (about 5 min). The final product was colorimetrically characterized by CIELa*b*.

### 2.8. Statistical Analysis

All trials were performed in triplicate, and results were expressed as the mean ± standard deviation. The mean comparisons were carried out using an analysis of variance (ANOVA) to find the effect of individual factors and their interaction on colorimetric parameters (*p* < 0.01). Tukey’s post-hoc test (HSD) was also performed (*p* < 0.05) using EXCEL^®^ extension DSAASTAT for multiple comparison between samples.

## 3. Results and Discussion

### 3.1. Plant Material and Enzyme Preparations

Spinach was characterized in terms of moisture, DM, pH, and composition of the cell wall polysaccharides (cellulose, hemicellulose, and pectin). The obtained results (Table 2) are in line with those reported in the literature [36,37]. The main structural polysaccharides were: cellulose 40% (5.20 ± 0.08 g/100 g_dwb_); hemicellulose 41% (5.35 ± 0.08 g/100 g_dwb_); and pectin 19% (2.43 ± 0.14 g/100 g_dwb_). On the basis of the spinach cell wall composition, it was possible to develop a tailored enzyme mix composed of 40% CL, 41% XL, and 19% PG.

Each enzyme preparation was initially characterized in terms of total protein content (0.024 mg_BSAeq_/mL for CL and PG, 0.027 mg_BSAeq_/mL for XL, Table 3). The SDS-PAGE profile (proved that all enzyme preparations were characterized by the presence of a dominant band which was in line with the previous reports [4].

The kinetic curves of the three enzyme preparations all followed the hyperbolic behavior of the Michaelis-Menten equation (Figure 1) and the estimated kinetic parameters were reported in Table 3.

### 3.2. Enzyme-Assisted Extraction for Chlorophyll Recovery

To maximize the amount of Chl recovered from spinach through an innovative eco-friendly EAE process, the applied conditions should be optimized since they have a strong influence on the amount of pigment extracted and its stability [38]. The effects of process variables (T, t, enzyme mix, Zn, B/S) on the amount of Chl extracted from spinach leaves were evaluated by RSM. Table 1 shows the results of the full factorial design, and Table 4 shows the coefficients of the mathematical model and statistical parameters, which display that the RSM developed was adequate (Model *p*-value = 0.003).

For the response variable considered (Chl concentration), the R^2^ value was equal to 0.93, indicating that the regression model was able to effectively explain how the variables considered and their interactions affected the response.

Furthermore, the significance of each equation coefficient was determined using the p-value (Table 4), and the smaller the p-value, the more significant the corresponding coefficient [39]. The analysis of variance was performed to determine the significance of the linear, quadratic, and interaction effects of the independent variable on the dependent variable. All the considered variables, except time, had a statistically significant linear effect (*p*-value < 0.05, Table 4). In detail, T (*p*-value < 0.01), Zn concentration (*p*-value < 0.01) and B/S (*p*-value < 0.001) were the factors that contributed the most (both positively and negatively) to the extraction of Chl when their linear effect was considered. The quadratic effect was not significant (*p*-value > 0.05) for any variable (Table 4). Otherwise, the interactions among the variables that most affected the model (*p*-value < 0.05) were T (°C) × Zn (ppm), time (h) × Enzyme mix (U/g), t (h) × Zn (ppm) and t (h) × B/S. It may be observed that time was significant only in terms of interaction with other variables. The results allowed us to develop a useful equation [40] to predict the amount of Chl extracted by varying T (x_1_), t (x_2_), enzyme mix (X_3_), Zn (X_4_) and B/S (X_5_) (Equation (4)).
Chl = +794.3 − 105.1x_1_ + 7.0x_2_ + 61.7x_3_ + 91.5x_4_ − 144.2x_5_ + 90.5x_1_^2^ + 41.9x_2_^2^ − 100.9x_3_^2^ + 99.0x_4_^2^ − 126.7x_5_^2^ − 22.6x_1_x_2_ − 18.5x_1_x_3_ + 71.1x_1_x_4_ + 31.4x_1_x_5_ − 101.2x_2_x_3_ + 95.6x_2_x_4_ + 81.2x_2_x_5_ + 59.9x_3_x_4_ − 47.5x_3_x_5_ − 55.8x_4_x_5_(4)

The interactions between the variables that most affected the response are depicted through the contour plots (Figure 2). Setting the values for T = 37.5 °C, Zn = 150 ppm and B/S = 17.5 (Figure 2a), it may be observed that the highest Chl concentration (>800 μg/g) was recovered at the highest enzymatic mix doses (>28 U/g), and also in the shortest extraction time (1 h). An extended extraction time appeared necessary in view of minimizing the enzymatic dose (up to 15 U/g) (Figure 2a), thus lowering the production costs. The effect of the interaction T (°C) × Zn (ppm) on the amount of Chl extracted is shown in Figure 2b (value set t = 3 h, Enzyme mix = 30 U/g and B/S = 17.5). It is interesting to point out that the highest concentration of Chl was reached at temperatures between 25 and 32 °C, independent of the amount of Zn used. By increasing the temperature (>32 °C), the protective action exerted by Zn on the green pigment became more evident. The effect of the interaction t (h) × Zn (ppm) clearly proved the protective action of Zn, notably at a longer extraction time (Figure 2c) (set conditions T = 37.5 °C, Enzyme mix = 30 U/g and B/S = 17.5).

The literature lacks studies concerning Chl recovery by enzymes. Özkan et al. [7] described the extraction of zinc-stabilized Chl from spinach using a commercial preparation (Pectinex Ultra SP-L enzyme). They identified three parameters that affect the extraction (enzyme preparation concentration, extraction time, and temperature), and in line with our results, observed a significant and positive effect of the enzyme dose and temperature variables. This same relationship between enzyme dose, extraction time, and temperature was also observed on Pandan leaves [28]. The optimal extraction conditions found by Özkan et al. [7] were: (i) enzyme concentration 8%, (ii) temperature 45 °C, (iii) time 120 min, and (iv) Zn dose 300 ppm. Moving beyond the state of the art, our research developed a tailored enzyme mix formulated on the base of polysaccharide composition of the spinach cell wall. The optimized process conditions for maximizing the Chl recovery are as follows: 25 °C, 1 h, 50 U/g of enzyme mix, 300 ppm of Zn and 17.5 B/S ratio. These results prove that Chl extraction could be a green route to coloring foodstuffs, giving much greater value to unsold spinach and developing an eco-friendly and high-performing extraction process.

### 3.3. Colorimetric Properties of the Extracts

In addition to the quantification of the extracted pigment, other parameters to be considered for a more comprehensive picture of the green color quality are the CIELa*b* parameters (L*, a*, b*) and color indices (C*, h°). L* represents brightness (0 to 100), a* the green/red component (−a* = green, +a* = red), and b* the blue/yellow component (−b* = blue, +b* = yellow). In particular, as reported in the literature [7,28], the assessment of greenness is expressed by the parameter a*.

The values of L*, a*, b*, h° and C* and the corresponding resulting color of the Chl-based extracts from spinach under different conditions are summarized in Table 5.

Regarding the parameter a*, the variables that seem to have the most influence are temperature and time. In detail, we may observe how the lowest values (a* < −30, nuance towards green) are obtained at the lowest temperature (25 °C) and extraction time (1 h). Under such mild conditions, the amount of Zn did not seem to be relevant, whereas its protective action became evident in the extracts recovered at higher temperatures and for longer extraction times (>37.5 °C and >1 h). As expected, the color saturation (C*) and hue (h°) parameters, resulting from the a* value, followed the same trend (Table 5).

### 3.4. Overlay Plot of Chlorophyll and a* Value

To identify the region where it is possible to maximize both the concentration of Chl (800 < Chl < 1300 μg/g) and the value of a* (−30 < a* < −44), their corresponding contour plots were superimposed in an overlay plot (Figure 3) at fixed T (25 °C) and B/S (17.5).

The white area in Figure 3 delimited the optimal conditions to simultaneously maximize Chl and a* within the imposed ranges, using a Zn concentration of 100 ppm (value set). It corresponded to: time < 2 h and an enzyme mix dose 12–45 U/g.

### 3.5. Application of Green Colorant in a Food Matrix

The Chl-based extract from spinach, recovered by applying the selected conditions suggested by the overlay plot (T: 25 °C, t: 1 h, Enzyme mix: 30 U/g, Zn: 150 ppm, and B/S: 17.5), was used in the formulation of a real food (Baghrir) at different doses (0, 1, 2.5, 5, 7.5, and 10% *w*/*w* flour basis). The pictures and the coordinates of the tristimulus values (CIELa*b*) in Table 6 suggest a significant variation of the colorimetric parameters L* and a* between the product without colorant (0% *w*/*w*) and the product at low concentrations (1% *w*/*w*).

Concerning a*, it varied significantly towards negative values (more intense green nuance) up to a Chl-based extract concentration equal to 5% *w*/*w*. Over this dose, the addition of increased amount of colorant did not affect the colorimetric parameters a* and b* (Table 6).

## 4. Conclusions

The present work reports a biotechnological tool developed for the green recovery of chlorophyll from spinach, to be used as a natural food colorant. The process was performed using a tailored enzymatic mix, designed considering the polysaccharide composition of spinach cell wall. Response surface methodology allowed us to narrow the area in which it is possible to identify the highest amount of recovered chlorophyll, and concurrently, to have the best quality of green (T: 25 °C, Zn: 150 ppm e B/S: 17.5, t: <2 h, and enzyme mix dose between 12 and 45 U/g). The chlorophyll-based extract was successfully applied to give different green colorations to a real food. The main drawback of this approach is the cost of the biocatalyst and the intrinsic instability of the extracted pigment. Therefore, further studies will be devoted to minimizing the amount of the enzyme mix and to investigating the stability of the novel colorant preparations under different storage conditions (such as temperature and light exposure).

## Figures and Tables

**Figure 1 foods-12-03440-f001:**
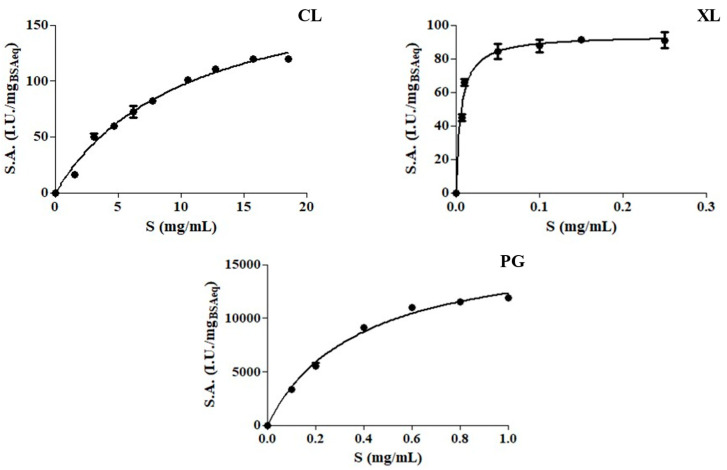
Michaelis–Menten plots of the enzyme preparations (cellulase (CL), xylanase (XL) and polygalacturonase (PG)). The specific activity (S.A., IU/mg_BSAEq_) were determined at 45 °C in McIlvaine buffer (0.1 M, pH 5), varying the substrate concentration (S, mg/mL).

**Figure 2 foods-12-03440-f002:**
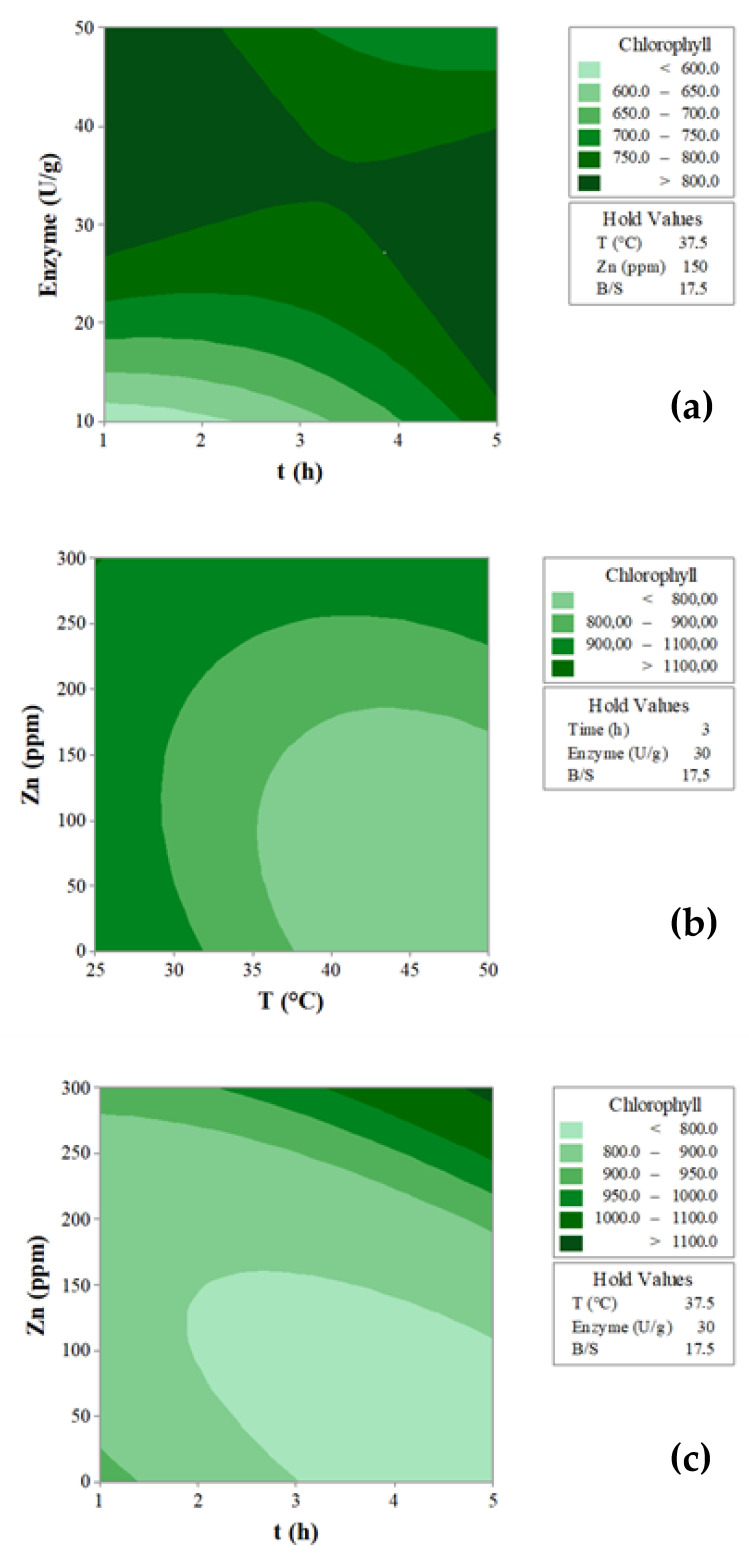
Contour plots showing the interactions between the variables with the greatest influence (*p* value ≤ 0.05) on the amount of chlorophyll extracted (μg/g): (**a**) time (h) × enzyme dose (U/g), (**b**) temperature (°C) × Zn (ppm), (**c**) time (h) × Zn (ppm).

**Figure 3 foods-12-03440-f003:**
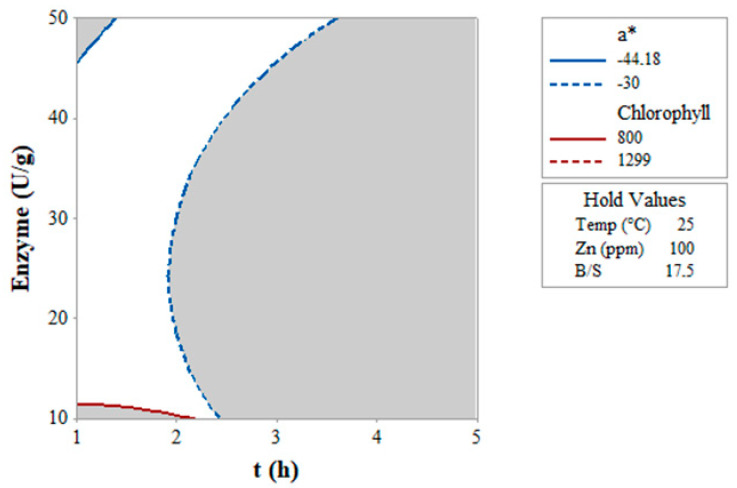
Overlay plot obtained by superimposing the contour plots of chlorophyll extracted (μg/g) and the quality of green (a*) as functions of enzyme dose (U/g) and time (t, h). The white area suggests the optimal region for maximizing both the amount of recovered chlorophyll and the quality of green color.

**Table 1 foods-12-03440-t001:** Design of Experiment (DOE) for the determination of optimal process parameters (temperature (°C), time (h), enzyme dose (enzyme mix, U/g), ZnCl_2_ dose (Zn, ppm), buffer/substrate ratio (B/S)) on the amount of chlorophyll extracted (Chl, μg/g).

Treatment Trial	Temperature (°C)	Time (h)	Enzyme Mix (U/g)	Zn (ppm)	B/S	Chl (mg/g)
	**X1 (x1)**	**X2 (x2)**	**X3 (x3)**	**X4 (x4)**	**X5 (x5)**	
1	37.5 (0)	3 (0)	30 (0)	150 (0)	17.5 (0)	791
2	25 (−1)	5 (+1)	50 (+1)	300 (+1)	5 (−1)	1299
3	25 (−1)	5 (+1)	10 (−1)	300 (+1)	30 (+1)	877
4	25 (−1)	1 (−1)	10 (−1)	300 (+1)	5 (−1)	783
5	25 (−1)	5 (+1)	50 (+1)	0 (−1)	30 (+1)	626
6	25 (−1)	1 (−1)	10 (−1)	0 (−1)	30 (+1)	597
7	50 (+1)	5 (+1)	50 (+1)	0 (−1)	5 (−1)	350
8	50 (+1)	1 (−1)	50 (+1)	300 (+1)	5 (−1)	1283
9	50 (+1)	5 (+1)	10 (−1)	300 (+1)	5 (−1)	993
10	50 (+1)	1 (−1)	50 (+1)	0 (−1)	30 (+1)	509
11	25 (−1)	5 (+1)	10 (−1)	0 (−1)	5 (−1)	869
12	50 (+1)	5 (+1)	50 (+1)	300 (+1)	30 (+1)	817
13	50 (+1)	5 (+1)	10 (−1)	0 (−1)	30 (+1)	590
14	50 (+1)	1 (−1)	10 (−1)	300 (+1)	30 (+1)	331
15	25 (−1)	1 (−1)	50 (+1)	300 (+1)	30 (+1)	636
16	25 (−1)	1 (−1)	50 (+1)	0 (−1)	5 (−1)	1286
17	37.5 (0)	3 (0)	30 (0)	150 (0)	17.5 (0)	791
18	50 (+1)	1 (−1)	10 (−1)	0 (−1)	5 (−1)	621
19	37.5 (0)	3 (0)	30 (0)	150 (0)	17.5 (0)	791
20	50 (+1)	3 (0)	30 (0)	150 (0)	17.5 (0)	718
21	37.5 (0)	5 (+1)	30 (0)	150 (0)	17.5 (0)	752
22	37.5 (0)	3 (0)	50 (+1)	150 (0)	17.5 (0)	716
23	37.5 (0)	3 (0)	30 (0)	150 (0)	17.5 (0)	714
24	37.5 (0)	3 (0)	30 (0)	300 (+1)	17.5 (0)	970
25	37.5 (0)	3 (0)	30 (0)	150 (0)	5 (−1)	755
26	25 (−1)	3 (0)	30 (0)	150 (0)	17.5 (0)	1130
27	37.5 (0)	3 (0)	30 (0)	150 (0)	30 (+1)	659
28	37.5 (0)	3 (0)	30 (0)	150 (0)	17.5 (0)	739
29	37.5 (0)	3 (0)	30 (0)	150 (0)	17.5 (0)	742
30	37.5 (0)	3 (0)	30 (0)	0 (−1)	17.5 (0)	895
31	37.5 (0)	1 (−1)	30 (0)	150 (0)	17.5 (0)	999
32	37.5 (0)	3 (0)	10 (−1)	150 (0)	17.5 (0)	750

**Table 2 foods-12-03440-t002:** Chemico-physical characterization of spinach.

Moisture (%)	Dry Matter (%)	pH	Cellulose (g/100 g_dwb_)	Hemicellulose (g/100 g_dwb_)	Pectin (g/100 g_dwb_)
93.6 ± 1.2	6.4 ± 0.9	6.29 ± 0.02	5.20 ± 0.08	5.35 ± 0.08	2.43 ± 0.14

dwb: dry weight basis.

**Table 3 foods-12-03440-t003:** Protein content and kinetic parameters (V_max_, K_M_, k_cat_, K_a_) of enzyme preparations (cellulase (CL), xylanase (XL) and polygalacturonase (PG)).

	Protein Content (mg_BSAeq_/mL)	V_max_ (I.U./mg_BSAeq_)	K_M_ (mg/mL)	k_cat_ (min^−1^)	Ka (min^−1^/mg mL^−1^)	R^2^
CL	0.024 ± 0.002	195.8 ± 9.94	10.41 ± 1.07	3.24 × 106 ± 9.94	3.11 × 105 ± 3.3 × 103	0.98
XL	0.027 ± 0.003	93.84 ± 2.02	0.0057 ± 0.0007	4.14 × 105 ± 2.02	7.29 × 107 ± 1.1 × 105	0.97
PG	0.024 ± 0.001	16950 ± 620	0.375 ± 0.034	6.13 × 108 ± 620	1.63 × 109 ± 1.3 × 107	0.99

V_max_: maximum rate at which the enzyme catalyses the reaction. K_M_: Michaelis-Menten constant. k_cat_ = V_max_/[E]_tot_, where [E]_tot_ is the molar concentration of the enzyme. K_a_ = k_cat_/K_M._

**Table 4 foods-12-03440-t004:** Regression coefficients, model *p*-value, R2 and adjusted R2 for the different polynomial models (Note: Subscripts: 0 = constant term; 1 = temperature; 2 = time; 3 = Enzyme dose; 4 = Zinc dose, 5 = buffer/substrate).

Regression Coefficient	Chl
b_0_	+794.3
b_1_	−105.1 **
b_2_	+7.0
b_3_	+61.7 *
b_4_	+91.5 **
b_5_	−144.2 ***
b_11_	+90.5
b_22_	+41.9
b_33_	−100.9
b_44_	+99.0
b_55_	−126.7
b_12_	−22.6
b_13_	−18.5
b_14_	+71.1 *
b_15_	+31.4
b_23_	−101.2 **
b_24_	+95.6 **
b_25_	+81.2 **
b_34_	+59.9
b_35_	−47.5
b_45_	−55.8
Model *p*-value	0.003
R^2^	0.93
R^2^ adj	0.77

* significant at 0.05 level, ** significant at 0.01 level, *** significant at 0.001 level.

**Table 5 foods-12-03440-t005:** Visual color attributes [a*, b*, L*, hue (h°), chroma value (C*)] of the chlorophyll-based extracts from spinach under different conditions.

Treatment Trial	L*	a*	b*	C*	h°	Color
1	47.14 ^f^	−20.26 ^i^	74.5 ^g^	77.2 ^hij^	105.2 ^ij^	
2	28.21 ^f^	−12.59 ^f^	47.2 ^p^	48.9 ^o^	104.9 ^ij^	
3	48.05 ^f^	−34.09 ^l^	73.4 ^gh^	80.9 ^fg^	114.9 ^l^	
4	65.07 ^c^	−26.68 ^j^	33.4 ^q^	42.7 ^p^	128.6 ^a^	
5	55.89 ^e^	−27.86 ^j^	84.1 ^d^	88.6 ^d^	108.3 ^gh^	
6	72.4 ^b^	−39.27 ^m^	95.5 ^b^	103.2 ^a^	112.4 ^e^	
7	42.75 ^gh^	−1.71 ^b^	54.62 ^o^	54.6 ^n^	91.8 ^n^	
8	23.29 ^j^	−14.11 ^f^	45.65 ^p^	47.8 ^o^	107.2 ^gh^	
9	30.95 ^i^	−3.48 ^bc^	51.92 ^o^	52.0 ^n^	93.8 ^mn^	
10	63.86 ^cd^	−26.72 ^j^	92.67 ^b^	96.4 ^bc^	106.1 ^jh^	
11	31.19 ^i^	−5.56 ^d^	52.72 ^o^	53.0 ^n^	96.0 ^l^	
12	38.96 ^g^	−0.18 ^a^	65.31 ^l^	65.3 ^m^	90.2 ^n^	
13	52.7 ^f^	−10.97 ^f^	82.75 ^d^	83.5 ^ef^	97.6 ^l^	
14	77.51 ^a^	−36.47 ^l^	98.42 ^a^	105.0 ^a^	110.3 ^fg^	
15	60.62 ^d^	−44.18 ^n^	85.24 ^d^	96.0 ^b^	117.4 ^d^	
16	40.74 ^g^	−43.38 ^n^	62.93 ^l^	76.4 ^ij^	124.6 ^b^	
17	48.14 ^f^	−19.26 ^i^	73.48 ^gh^	76.0 ^ij^	104.7 ^j^	
18	58.51 ^d^	−27.68 ^j^	87.73 ^c^	92.0 ^c^	107.5 ^hi^	
19	50.17 ^f^	−20.98 ^i^	71.22 ^ijk^	74.2 ^j^	106.4 ^hi^	
20	42.86 ^i^	−3.32 ^bc^	70.53 ^k^	70.6 ^k^	92.7 ^mn^	
21	43.86 ^g^	−5.78 ^cd^	71.35 ^jk^	71.6 ^k^	94.6 ^m^	
22	49.32 ^g^	−17.03 ^h^	77.75 ^f^	79.6 ^fjk^	102.4 ^j^	
23	49.25 ^f^	−15.8 ^gh^	77.67 ^f^	79.3 ^fjk^	101.5 ^j^	
24	39.1 ^f^	−14.25 ^f^	64.26 ^l^	65.8 ^l^	102.5 ^j^	
25	46.74 ^gh^	−19.72 ^i^	72.94 ^hi^	75.6 ^j^	105.1 ^j^	
26	35.78 ^f^	−30.49 ^k^	57.01 ^n^	64.7 ^l^	118.1 ^d^	
27	53.07 ^h^	−19.13 ^i^	82.32 ^de^	84.5 ^e^	103.1 ^j^	
28	48.98 ^f^	−16.02 ^gh^	76.99 ^fg^	78.6 ^ghi^	101.8 ^k^	
29	47.22 ^f^	−14.99 ^g^	76.78 ^fg^	78.2 ^ghi^	101.0 ^k^	
30	40.36 ^f^	−11.6 ^e^	66.47 ^l^	67.5 ^l^	99.9 ^k^	
31	38.77 ^f^	−37.51 ^m^	60.88 ^m^	71.5 ^k^	121.6 ^c^	
32	52.76 ^g^	−31.77 ^k^	79.15 ^e^	85.3 ^de^	111.9 ^f^	

C* [Chroma = (a*^2^ + b*^2^)^0.5^]; h° [hue angle = 180° + (arctan b*/a*)]. Values with different small letters (a–q) differ significantly (Tukey’s test, *p* = 0.05) according to treatment trial.

**Table 6 foods-12-03440-t006:** Visual color attributes (L*, a*, b*) of the real food (Baghrir) prepared by adding different amounts of chlorophyll-based extracts from spinach (0–10% *w*/*w* flour basis).

Colorant (%)	L*	a*	b*	Sample
0%	38.4 ^a^	2.64 ^a^	18.16 ^a^	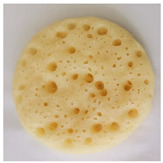
1.0%	36.43 ^b^	−1.53 ^b^	18.98 ^a^	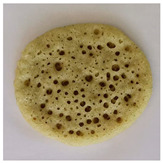
2.5%	23.94 ^d^	−2.09 ^c^	15.99 ^b^	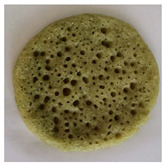
5.0%	22.01 ^e^	−3.92 ^d^	13.83 ^c^	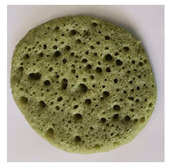
7.5%	26.75 ^c^	−3.98 ^d^	13.93 ^c^	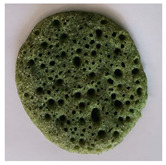
10%	26.15 ^c^	−4.41 ^d^	11.37 ^c^	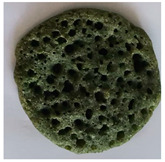

Values with different small letters (a–e) differ significantly (Tukey’s test, *p* = 0.05) according to colorant (%).

## Data Availability

The data presented in this study are available on request from the corresponding author.

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
