# Peer review of "Enzyme-Assisted Extraction for the Recovery of Food-Grade Chlorophyll-Based Green Colorant"

_foods, 2023, doi:10.3390/foods12183440_

Round 1
Reviewer 1 Report
In this study, the conditions for solvent-free recovery of chlorophylls, which can be obtained from unused vegetables, have been optimized to use as natural colorants instead of synthetic pigments. In this context, the amount of chlorophyll and the quality of the green color were taken into consideration. Additionally, spinach has been characterized in terms of moisture, DM, pH, and cell wall polysaccharide composition. The study is quite original. Experiments have been designed and conducted in detail. I will only have minor comments for some parts: When looking at the title of the study, it does not seem to be consistent with the content. Therefore, I suggest revising it. The introduction of the study started quite well. However, not mentioning the studies on obtaining chlorophyll from spinach and its application in food is a deficiency. A general and current literature summary should be given, and how this study differs from others and its originality should be stated.
The materials and methods sections are clear. The results are discussed well and the conclusion part is enough.
Author Response
Reviewer 1
In this study, the conditions for solvent-free recovery of chlorophylls, which can be obtained from unused vegetables, have been optimized to use as natural colorants instead of synthetic pigments. In this context, the amount of chlorophyll and the quality of the green color were taken into consideration. Additionally, spinach has been characterized in terms of moisture, DM, pH, and cell wall polysaccharide composition. The study is quite original. Experiments have been designed and conducted in detail. I will only have minor comments for some parts: When looking at the title of the study, it does not seem to be consistent with the content. Therefore, I suggest revising it. The introduction of the study started quite well. However, not mentioning the studies on obtaining chlorophyll from spinach and its application in food is a deficiency. A general and current literature summary should be given, and how this study differs from others and its originality should be stated.
The materials and methods sections are clear. The results are discussed well and the conclusion part is enough.
- We would like to thank the Reviewer for her/his observations. As suggested, the title has been modified and studies on obtaining chlorophyll from spinach have been added in the introduction. Therefore, in the aim, it is clearly indicated how this study differs from others and its originality.
Reviewer 2 Report
Overall, your manuscript has the potential to provide valuable insights into the solvent-free recovery of chlorophyll from spinach for food applications. However, there are some suggestions and comments to consider for improvement:
- Title: The title can be made more concise and specific. For example, you could mention the use of enzyme-assisted extraction or emphasize the circular economy aspect. A more concise title might attract more readers.
- Abstract: The abstract is well-structured and informative. However, it could be made more concise. Try to summarize the main objectives, methods, and key findings in a more succinct manner.
- Introduction:
- Consider providing a brief overview of the circular economy model and its relevance to the food industry earlier in the introduction.
- Define acronyms like "EAE" (Enzyme-assisted Extraction) when they are first introduced.
- Materials and Methods:
- Provide more details on the specific enzyme preparations used, such as enzyme concentrations and sources.
- In the design of experiments (DOE) section, explain the rationale behind the chosen ranges for each variable. Why were those specific values selected?
- Clarify the units of measurement used throughout the methods section. For example, specify the units for "Vmax" and "KM."
- Consider providing information on the pH of the McIlvaine buffer used, as this can be relevant to enzyme activity.
- In the application of chlorophyll-based colorant in a food matrix, consider including sensory evaluation results or comments on the taste and texture of the Baghrir prepared with different concentrations of colorant. This would provide a more comprehensive view of the impact of the colorant on the final product.
- Results and Discussion:
- Ensure that the results and discussion sections are clearly separated.
- When discussing the results of the extraction process optimization, provide more context on why certain conditions were chosen. Explain why those specific conditions are considered optimal.
- Use subheadings or bullet points to make the discussion section more organized and easier to follow.
- Discuss the practical implications of your findings in the context of the food industry. How could the optimized extraction process be applied in real-world food production?
- Figures and Tables:
- Include figure captions for clarity. Explain what each figure represents and its significance.
- Ensure that figures and tables are referenced in the text and vice versa.
- Language and Clarity:
- Proofread the manuscript for language and grammar errors.
- Use consistent terminology and units of measurement throughout the manuscript.
- Consider rephrasing complex sentences to improve readability.
- References:
- Make sure all references are properly formatted according to the chosen style guide (e.g., APA, Chicago).
- Conclusion:
- Summarize the key findings and their potential impact on the food industry.
- Discuss any limitations of the study and suggest directions for future research.
- Overall Organization:
- Review the overall organization of the manuscript to ensure a logical flow from introduction to conclusion.
By addressing these suggestions, you can enhance the clarity and impact of your manuscript for potential readers and researchers in the field.
minor issues
Author Response
Reviewer 2
Overall, your manuscript has the potential to provide valuable insights into the solvent-free recovery of chlorophyll from spinach for food applications. However, there are some suggestions and comments to consider for improvement:
- Title: The title can be made more concise and specific. For example, you could mention the use of enzyme-assisted extraction or emphasize the circular economy aspect. A more concise title might attract more readers.
- According to Reviewer’s suggestion, the title has been changed.
- Abstract: The abstract is well-structured and informative. However, it could be made more concise. Try to summarize the main objectives, methods, and key findings in a more succinct manner.
- Accordingly to Reviewer’s observation, the abstract has been summarized.
- Introduction:
Consider providing a brief overview of the circular economy model and its relevance to the food industry earlier in the introduction.
- Following the Reviewer’s suggestions, a brief overview of the circular economy model and its relevance to the food industry has been added in the introduction.
Define acronyms like "EAE" (Enzyme-assisted Extraction) when they are first introduced.
- The define of acronyms EAE has been added.
- Materials and Methods:
Provide more details on the specific enzyme preparations used, such as enzyme concentrations and sources.
- More details on the specific enzyme preparations used in this study have been reported.
In the design of experiments (DOE) section, explain the rationale behind the chosen ranges for each variable. Why were those specific values selected?
- The range of values ​​selected for each variable was chosen as a compromise between the conditions that are most commonly reported in literature for pigment extraction, and which may make the process more eco-sustainable and easily applicable in the food industry.
Clarify the units of measurement used throughout the methods section. For example, specify the units for "Vmax" and "KM."
- All the unit of measurement used throughout the methods section have been revised and added.
Consider providing information on the pH of the McIlvaine buffer used, as this can be relevant to enzyme activity.
- As reported in section 2.5 (Chlorophyll recovery by Enzyme Assisted Extraction), the pH of the McIlvaine buffer is equal to 5.
In the application of chlorophyll-based colorant in a food matrix, consider including sensory evaluation results or comments on the taste and texture of the Baghrir prepared with different concentrations of colorant. This would provide a more comprehensive view of the impact of the colorant on the final product.
- We would like to thank the Reviewer for her/his observation. A preliminary sensory evaluation was carried out by our laboratory staff and gave a positive result; no difference was revealed using a concentration up to 5% Chl in comparison with the control sample. However, since the quantity of available Chl extract was exiguous to perform a comprehensive sensory evaluation by a large number of panellists, this aspect was not included so far. Further studies will be performed with the aim of testing the sensory impact of Chl colorant recovered by EAE incorporated in different real foods and beverages.
- Results and Discussion:
Ensure that the results and discussion sections are clearly separated. When discussing the results of the extraction process optimization, provide more context on why certain conditions were chosen. Explain why those specific conditions are considered optimal. Use subheadings or bullet points to make the discussion section more organized and easier to follow.
- The paragraph 3.2 has been revised.
Discuss the practical implications of your findings in the context of the food industry. How could the optimized extraction process be applied in real-world food production?
- For the consumer and the food industry, it may be useful to have a colorant free of organic solvent residues, stabilized with Zn2+ (which is less toxic than Cu2+), and produced by a more eco-sustainable extraction approach.
- Figures and Tables:
Include figure captions for clarity. Explain what each figure represents and its significance. Ensure that figures and tables are referenced in the text and vice versa.
- The entire manuscript has been double-checked. The captions of all figures and tables are reported correctly and all are cited in the text and vice versa.
- Language and Clarity:
Proofread the manuscript for language and grammar errors. Use consistent terminology and units of measurement throughout the manuscript. Consider rephrasing complex sentences to improve readability.
- The entire manuscript has been double-checked.
- References:
Make sure all references are properly formatted according to the chosen style guide (e.g., APA, Chicago).
- All references have been double-checked.
- Conclusion:
Summarize the key findings and their potential impact on the food industry. Discuss any limitations of the study and suggest directions for future research.
- Following the Reviewer’s suggestions, the conclusion section has been revised.
- Overall Organization:
Review the overall organization of the manuscript to ensure a logical flow from introduction to conclusion.
- The overall organization of the manuscript has been reviewed.
By addressing these suggestions, you can enhance the clarity and impact of your manuscript for potential readers and researchers in the field.
Reviewer 3 Report
The article “Solvent-free recovery of chlorophyll-based green colorant from spinach for food application” by Mazzocchi et al. describe enzymatic assisted preparation of chlorophyll containing fraction from unused spinach leaves. The authors achieved separation of green dye and use it for coloring of a traditional biscuit during backing. Moreover, they characterized the starting spinach material.
The equation (1) can be simplified to DM = dw/fw * 100
Instead of “3’5’-dinitro…” should be “3’,5’-dinitro…”
Could authors provide SDS-PAGE profiles of enzyme preparation?
Which role plays the difference of absorbtion of Zn-Chl and Mg-Chl on color of prepared food?
Author Response
Reviewer 3
The article “Solvent-free recovery of chlorophyll-based green colorant from spinach for food application” by Mazzocchi et al. describe enzymatic assisted preparation of chlorophyll containing fraction from unused spinach leaves. The authors achieved separation of green dye and use it for coloring of a traditional biscuit during backing. Moreover, they characterized the starting spinach material.
The equation (1) can be simplified to DM = dw/fw * 100
- Done
Instead of “3’5’-dinitro…” should be “3’,5’-dinitro…”
- Done
Could authors provide SDS-PAGE profiles of enzyme preparation?
- Electrophoretic profile (SDS-PAGE) of the enzyme preparations used [cellulase (CL), xylanase (XL) and polygalacturonase (PG)]. The relative molecular mass of the protein standards (ST, kD) is indicated in the first column (Please see the attachment).
Which role plays the difference of absorbtion of Zn-Chl and Mg-Chl on color of prepared food?
- Factors (e. g. oxygen, pH, light, temperature, and chlorophyllase) can cause a centralized shift of the Mg ion in the central position, giving rise to pheophytin, which causes a color change towards an olive green/gray tone. It is possible to produce more stable complexes by substituting Mg2+ with Zn2+, thus re-storing the functions and properties of the molecule, producing a greening effect. No absorption differences have been highlighted on color of prepared food.
